# Using Language to Extend to Unseen Domains

**Lisa Dunlap, Clara Mohri** *
UC Berkeley
{lisabdunlap,cmohri}@berkeley.edu

**Aditi Raghunathan**
Carnegie Mellon University
raditi@cmu.edu

**Han Zhang, Devin Guillory, Trevor Darrell, Joseph E. Gonzalez, Anna Rohrbach**
UC Berkeley
{pariszhang,dguillory,trevordarrell,jegonzal,anna.rohrbach}@berkeley.edu

## Abstract

It is expensive to collect training data for every possible domain that a vision model may encounter when deployed. We instead consider how simply *verbalizing* the training domain (e.g. "photos of birds") as well as domains we want to extend to but do not have data for (e.g. "paintings of birds") can improve robustness. Using a multimodal model with a joint image and language embedding space, our method *LADS* learns a transformation of the image embeddings from the training domain to each unseen test domain, while preserving task relevant information. Without using any images from the unseen test domain, we show that over the *extended* domain containing both training and unseen test domains, *LADS* outperforms standard fine-tuning and ensemble approaches over a suite of four benchmarks targeting domain adaptation and dataset bias. Code is available at https://github.com/lisadunlap/LADS.

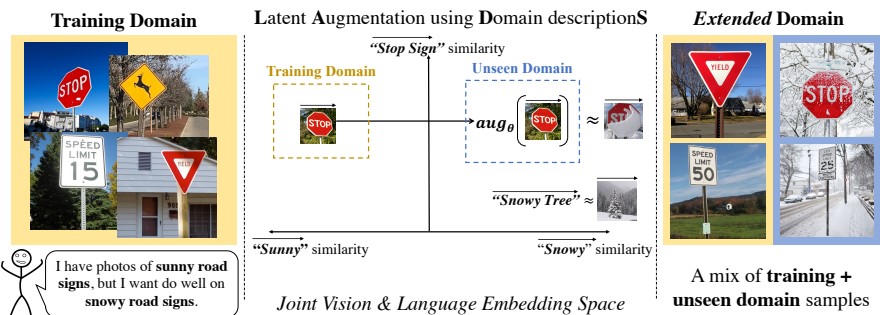

Figure 1: Consider a model trained to recognize road signs in sunny weather. We aim to *extend* to a new domain of snowy weather. Our method *LADS* (Latent Augmentation using Domain descriptionS) leverages a multimodal model's knowledge of the classes and the domain shift verbalized in natural language ("sunny" to "snowy") to train an augmentation network without any samples from the unseen test domain. This network is used to translate multimodal *image embeddings* from the training domain to the unseen test domain, while retaining class-relevant information. Then, real and augmented embeddings are used jointly to train a classifier.

## 1 Introduction

The ability to extend a model beyond the domain of the training data is central to building robust computer vision models. Methods for dealing with unseen test distributions often require leveraging additional image data, but linguistic knowledge of the anticipated domain shift is much cheaper and easier to obtain. For example, in many settings, the training images are collected in certain

---

*equal contribution

conditions (e.g., daylight, clear weather, ...) but our sensors may also experience less common but easy to anticipate conditions (e.g., night, snow, haze, illustrations, ...). Directly collecting or creating data in all possible anticipated settings is often prohibitively expensive. Thus it is of great interest how one can *linguistically extend* to unseen domains: that is, to utilize language to improve performance on an unseen test domain without sacrificing performance on the training domain.

The use of language in domain generalization has generated significant interest with the development of large vision-language models such as CLIP (Radford et al., 2021), Flamingo (Alayrac et al., 2022), and ALIGN (Jia et al., 2021), which allow users to create zero-shot classifiers using only class names. However, while these models have been shown to achieve remarkable cross-domain generalization, their zero-shot classifiers often perform far worse than models trained for a particular downstream task (Radford et al., 2021; Kumar et al., 2022). When training data is available for the downstream task, a common practice is to fine-tune these models on the training data. While this significantly improves in-domain accuracy, it degrades performance on unseen domains.

We show that it is possible to leverage the domain-level knowledge (e.g. sunny environments vs. snowy environments in our example) contained in CLIP or similar models to deal with a variety of domain shifts in a way that requires no data from the new test domain, exploits the labeled training data, and is fast to train. Our method only requires users to input text descriptions of the training and unseen test domains (e.g. "a sunny stop sign" and "a snowy stop sign") along with their training data. To achieve language-guided domain generalization, we leverage the broad domain knowledge encoded in CLIP coupled with its shared image-language embedding space to perform *latent feature augmentation* of the training set.

More precisely, the embeddings of these textual descriptions are used to train an augmentation model which learns a transformation on the CLIP image embeddings of the training domain and "places" them in the new domain (see Figure 1). We train this augmentation model with two objectives: (1) translating the image embedding from the training domain to the unseen testing domain, while (2) retaining the class-specific information of the original image. Once this transformation is learned, we train a simple linear classifier on the combined augmented and unaugmented image embeddings, resulting in a classifier that outperforms common fine-tuning methods on the extended domain while achieving similar performance on the training domain.

We introduce *LADS*, a method to extend a model to new domains given only a language description of the distribution shift. Our main contributions are (1) the introduction of the *Domain Extension with Language* problem, (2) a novel language-guided *latent feature augmentation* training procedure, and (3) the extension of our method to address spurious correlation biases in the training data.

We evaluate *LADS* on two domain adaptation benchmarks, DomainNet (Peng et al., 2019) and CUB-Paintings (Wang et al., 2020), as well as two benchmarks exhibiting color and contextual bias, Colored MNIST (Arjovsky et al., 2021) and Waterbirds (Sagawa et al., 2019). On the domain adaptation benchmarks, we show that we improve out-of-domain performance by 1-3% while matching in-domain performance of fine-tuned and ensembled models. On the biased benchmarks, we show an almost 2x improvement in out-of-domain performance over fine-tuned models. Across all benchmarks, *LADS* achieves the highest accuracy on the entire extended test domain containing both training and unseen test domain samples. Finally, we perform an in-depth analysis of the altered image embeddings, the effect of each loss function, and the effect of different vision and language models to understand our framework better.

## 2    RELATED WORK

**Domain Adaptation/Generalization.** The challenge of out-of-domain generalization is well studied (Recht et al., 2019; Petryk et al., 2022; Kumar et al., 2022; Santurkar et al., 2021; Hendrycks & Dietterich, 2019) with a large body of work in domain adaptation addressing the problem of adapting a model to perform well on a new target domain. A typical domain adaptation approach involves collecting additional unlabeled data from the target domain (Ganin & Lempitsky, 2015; Saito et al., 2017; Arjovsky et al., 2021; Kim et al., 2018; Tzeng et al., 2015), and aims to train a classifier such that it cannot tell the difference between source and target domain.

In the limited data setting, few-shot domain adaptation (Motiian et al., 2017; Yue et al., 2021) aims to learn from as little as one example in the target domain. Work in domain generalization (Wang &

Jiang, 2020; Gulrajani & Lopez-Paz, 2021; Koh et al., 2021) does not need target domain data but requires a set of several aligned and labeled source domains, and often shows only limited gains. While we evaluate on certain domain adaptation benchmarks, DA/DG methods primarily focus on maximizing target domain accuracy, while our work is interested in maximizing the accuracy of the *extended domain*. Furthermore, unlike previous works, we assume we have no access to any target data (labeled or unlabeled), only a single source domain, and our domain shift can be verbalized.

**Fine-tuning under Distribution Shift.** The goal of fine-tuning under distribution shift is to tailor pretrained models to a specific task without sacrificing their ability to deal with distribution shifts. Kumar et al. (2022) found that it is better to fit a linear probe on the features and then fine-tune the model's backbone. For robust fine-tuning of CLIP specifically, Wortsman et al. (2021) proposed ensembling the weights of the fine-tuned image encoder with the zero-shot image encoder. We see our work as complementary to these ideas, targeting semantically defined domain shifts to increase OOD performance, while maintaining high ID performance.

**Semantic Augmentation with CLIP.** With the emergence of CLIP, several works (Ramesh et al., 2022; Patashnik et al., 2021; Gal et al., 2021) have used language to alter images using a combination of CLIP and a generative model. Broadly, these works translate an image to a CLIP embedding, alter the image embedding with a text embedding of the desired augmentation, and use that embedding to generate an altered image. These CLIP-based works do not attempt to use these data augmentations in the context of dataset bias or domain adaptation. Some prior work has explored augmentations using generative models (Sharmanska et al., 2020; Sankaranarayanan et al., 2018; Yan et al., 2021), but since they generate images at the pixel level, they are often bottle-necked by the quality of the generative process. In contrast, we choose to manipulate embeddings directly that allows us to effectively distill the knowledge in CLIP.

**Removing Dataset Bias.** In computer vision, several works debias data using extra information such as instance annotations (Hendricks et al., 2018; Li et al., 2018; Rieger et al., 2020), bounding boxes (Choi et al., 2019), or image-level bias annotations (Kim et al., 2018). Some methods (Sharmanska et al., 2020; Bau et al., 2020; Santurkar et al., 2021) forego the need for expensive annotations by utilizing generative models, while Petryk et al. (2022) utilize CLIP to translate language descriptions of a task into spatial guidance. In contrast, we do not limit ourselves to purely spatial bias or use per-image annotations of the bias, only a description of what biases may appear in the training data.

## 3  Latent Augmentation using Domain Descriptions

We consider the supervised learning problem of generalizing to new unseen domains using only the verbal descriptions of the training domain and the anticipated but unseen new domains. More formally, we are given a training dataset $\{\mathbf{x_i}, y_i\}_{i=1}^n$ drawn from the training domain $D_{\text{training}}$, the class names $\mathbf{t}_y$, a written description $t_{\text{training}}$ of the training domain, and a set of written descriptions $\{t_{\text{unseen}}^i\}_{i=1}^k$ of $k$ unseen domains $\{D_{\text{unseen}}^i\}_{i=1}^k$ that we expect to encounter at test time. Our goal is to train a model that performs well on both the original domain $D_{\text{training}}$ as well as the unseen domains $\{D_{\text{unseen}}^i\}_{i=1}^k$. We call this the *Domain Extension with Language* problem.

Large vision-language models have demonstrated the ability to generalize to new domains with language but only in the zero-shot setting. In order to utilize available training data, we explore the popular fine-tuning technique of *linear probing*: fitting a linear classifier to the image embeddings of large vision-language models. We chose linear probing over full fine-tuning as it is faster to train and has been shown to result in more robust classifiers (Kumar et al., 2022; Radford et al., 2021).

While standard linear probing only uses the image embeddings and the numerical labels, *LADS* also utilizes the text describing the classes and the descriptions of domain shift to augment the probe's training data to mimic samples from the unseen domain. Our two-stage approach first learns a network that transforms the *image embeddings* rather than the pixels themselves, with the goals of (1) augmenting the embedding to be aligned with the unseen domain while (2) retaining the features consistent with its class label. The second stage performs linear probing on the training set containing both the original image embeddings as well as the augmented image embeddings to produce a classifier that is more robust to the specified domains. Note that we do not use any data from $D_{\text{unseen}}^k$ in either stage—we only use the class names and domain descriptions. An outline of the first stage of our method (training the augmentation network) is depicted in Figure 2.

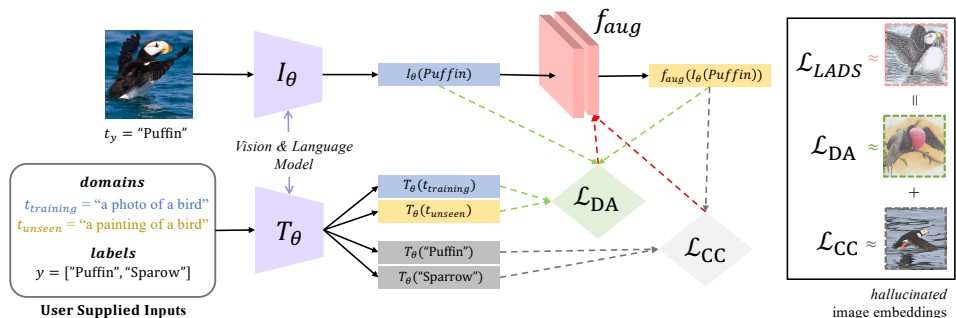

Figure 2: **LADS**. Let the task be to classify *Puffin* vs. *Sparrow*. The training data $D_{\text{training}}$ contains *photos* of the two classes but we would like to extend our classifier to *paintings* as well: that is, $D_{\text{unseen}}$. We aim to do this using the text descriptions of the training and new domain, $t_{\text{training}}$ and $t_{\text{unseen}}$, respectively. The augmentation network $f_{\text{aug}}$ is trained to transform image embeddings from $D_{\text{training}}$ to $D_{\text{unseen}}$ using a *domain alignment* loss $\mathcal{L}_{\text{DA}}$ and a *class consistency* loss $\mathcal{L}_{\text{CC}}$. When $\mathcal{L}_{\text{DA}}$ is low, the augmented embeddings are in the new domain but may have drifted from their class. When $\mathcal{L}_{\text{CC}}$ is low, the augmented embeddings will retain class information but may fail to reflect the desired change in domain. $f_{\text{aug}}$ aims to augment every image embedding to a space with low domain alignment loss *and* low class consistency loss, resulting in $f_{\text{aug}}(I(\mathbf{x}))$ having an image embedding similar to a painting of a Puffin. Note that the hallucinated image embeddings on the right are a pictorial representation of the effect of each loss function and not actually generated by *LADS*.

We choose CLIP (Radford et al., 2021) as our vision-language model in our evaluations. Let $I_\theta(\mathbf{x}) = \text{CLIP}_{\text{img}}(\mathbf{x}) \in \mathcal{I}$ denote the image embedding of input image $\mathbf{x}$ and $T_\theta(t) = \text{CLIP}_{\text{text}}(t) \in \mathcal{T}$ denote the CLIP text embedding of some text $t$. Furthermore, let $t_{\text{training}} \circ \text{t}_y$ denote the composition of the domain description and the class name. For example, if $t_{\text{training}} = $ *"a photo of a"*, $t_{\text{unseen}}^1 = $ *"a painting of a"* and $\text{t}_y$ could be *"Puffin"*. The composition $t_{\text{training}} \circ \text{t}_y$ is *"a photo of a Puffin"*.

**Stage 1: Training the augmentation network.** The first stage of *LADS* is to learn an augmentation network $f_{\text{aug}}^k : \mathcal{I} \to \mathcal{I}$ that transforms image embeddings from $D_{\text{training}}$ to $D_{\text{unseen}}^k$ using the corresponding language descriptions $t_{\text{training}}$ and $t_{\text{unseen}}^k$. As mentioned previously, a valuable augmentation is one which places the transformed embedding in unseen domain $D_{\text{unseen}}^k$ while retaining features relevant to the class label. To achieve this, we train $f_{\text{aug}}^k$ using a combination of two losses: *Domain Alignment* and *Class Consistency*. In the setting of adapting to multiple new domains at once, we train a unique $f_{\text{aug}}^k$ network for each domain as described above.

*Domain Alignment.* The domain alignment loss encourages the augmented image embeddings $f_{\text{aug}}^k(I_\theta(\mathbf{x}))$ to look like image embeddings from the new domain $D_{\text{unseen}}^k$. This loss is guided by the text embeddings of the domain descriptions $t_{\text{unseen}}^k$ and $t_{\text{training}}$.

While CLIP is trained such that the space of image embeddings $\mathcal{I}$ has some correspondence with the space of text embeddings $\mathcal{T}$, it is not obvious what a mapping between $\mathcal{I}$ and $\mathcal{T}$ should look like. Thus, inspired by prior work (Patashnik et al., 2021; Gal et al., 2021), we assume the existence of a "global direction" that corresponds to a shift from $D_{\text{training}}$ to $D_{\text{unseen}}^k$ that is shared across both the image embedding space and text embeddings space.

This "global direction" is defined as the normalized difference of the embeddings from the target domain and the embeddings from the source domain. Formally, the domain alignment loss of $f_{\text{aug}}^k$ for training point $(\mathbf{x_i}, y_i)$ is

$$\mathcal{L}_{\text{DA}}(f_{\text{aug}}^k) = \sum_{i=1}^{n} 1 - \left( \frac{f_{\text{aug}}^k(I_\theta(\mathbf{x_i})) - I_\theta(\mathbf{x_i})}{\|f_{\text{aug}}^k(I_\theta(\mathbf{x_i})) - I_\theta(\mathbf{x_i})\|} \cdot \frac{T_\theta(t_{\text{unseen}}, y_i) - T_\theta(t_{\text{training}}, y_i)}{\|T_\theta(t_{\text{unseen}}, y_i) - T_\theta(t_{\text{training}}, y_i)\|} \right). \quad (1)$$

*Class Consistency.* The domain alignment loss in Equation 1 encourages the augmented embeddings to only differ in the direction of change in the domain. If there were one global shared direction corresponding to the domain shift, optimizing $\mathcal{L}_{\text{DA}}$ would be sufficient. However, in practice, we

find that optimizing $\mathcal{L}_{\text{DA}}$ alone removes some class relevant information and results in little diversity among the augmented embeddings of different images (see Section 4.6 and Figure 11). Thus we add a class consistency loss which preserves class information in the augmented embeddings. We measure class information in the image embeddings by our ability to classify the images accurately via CLIP zero-shot with the class names. Formally,

$$\mathcal{L}_{\text{CC}}(f_{\text{aug}}^k) = \sum_{i=1}^{n} \text{Cross-entropy}\big(\text{Softmax}[f_{\text{aug}}^k(I_\theta(\mathbf{x_i})) \cdot T_\theta(y_i)], y_i\big) \qquad (2)$$

Note that this is the same objective as the standard CLIP loss. We use CLIP zero-shot rather than trying to fine-tune CLIP because that could lead to overfitting where we classify the augmented training image embeddings correctly even when they do not contain class relevant information.

Our final objective $\mathcal{L}_{\text{LADS}}(f_{\text{aug}})$ to train the augmentation network as the first step in LADS is a linear combination of the the domain alignment loss and class consistency loss:

$$\mathcal{L}_{\text{LADS}}(f_{\text{aug}}^k) = \alpha\mathcal{L}_{\text{DA}}(f_{\text{aug}}^k) + (1-\alpha)\mathcal{L}_{\text{CC}}(f_{\text{aug}}^k), \qquad (3)$$

where $\alpha$ is a hyperparameter dictating the trade-off between domain alignment and class consistent.

**Stage 2: Fine-tuning.** After the augmentation network $f_{\text{aug}}^k$ is trained, we train a linear probe on the original image embeddings $I_\theta(\mathbf{x_i})$ along with the augmented embeddings $f_{\text{aug}}^k(I_\theta(\mathbf{x_i}))$. Inference is straightforward: apply the linear probe on the CLIP image embeddings of the test images.

## 3.1 Addressing dataset bias

In addition to dealing with extended domains, *LADS* can also be used in the dataset bias setting where there are spurious correlations in the dataset. For example, in Waterbirds (Sagawa et al., 2019), we want to classify Landbirds vs. Waterbirds, where the spurious correlation is the background (Landbirds appear on forest backgrounds and Waterbirds appear on water backgrounds in training). To prevent a classifier from using this correlation to make predictions, we can use *LADS* to generate augmentations that represent "Landbird on water" and "Waterbird on land".

We do this by using CLIP to label the backgrounds of each image and then decide what $t_{\text{training}}$ and $t_{\text{unseen}}$ is per example. Given the domain information $t_{land}$ = "a {} in the forest" and $t_{water}$ = "a {} on the water", we can use zero-shot CLIP to determine if a given image is on land or water. If the image is predicted to be on land, when training $f_{\text{aug}}$, $\mathcal{L}_{\text{DA}}$ for that particular example will use $t_{\text{training}} = t_{land}, t_{\text{unseen}} = t_{water}$ and vice versa. The class consistency loss and the other parts of the pipeline remain unchanged. Because we are using the vision and language model to label the domains, we do not need per-image labels of the bias, only a hypothesis of what the bias may be.

## 4 Experiments

In this section we discuss our main experiments and results. We defer dataset details, the remainder of the experiments and their discussion to the Appendix (B, D, E).

## 4.1 Implementation Details

In line with Radford et al. (2021), we normalize all text and image embeddings when performing zero-shot inference or training with CLIP embeddings. The augmentation network $f_{\text{aug}}$ used in *LADS* is a 2-layer MLP with input and output dimensions of 768 and a hidden dimension of 384. Within *LADS* and all the CLIP-related baselines, we use the OpenAI CLIP model with a ViT-L backbone and resize all images to 224x224. We train on 10 GeForce RTX 2080 Ti GPUs.

For each baseline, we do a hyperparameter sweep across learning rate and weight decay and choose the parameters with the highest class-balanced validation accuracy. For *LADS* we also do a sweep across the parameters of the augmentation network, namely learning rate, weight decay, and $\alpha$, and

select a checkpoint based on the validation loss. In general, we set $\alpha = 0.5, lr = 0.001, wd = 0.05$. Our hyperparameter search spaces and final choice of hyperparameters are listed in Table 4.

In our results we report test accuracy on $D_{\text{training}}$, $D_{\text{unseen}}$, and the extended domain which averages the two. We run each method over 5 different random seeds and report the mean and standard deviation.

## 4.2 DATASETS

**CUB-Paintings (one new domain)** is composed of 2 datasets, CUB-200 (Wah et al., 2011), a fine-grained bird classification benchmark containing 200 different bird species and CUB-200-Paintings (Wang et al., 2020), which contains the same classes as CUB-200 but instead of real images they are paintings collected from the web and filtered manually. We use the domain descriptions $t_{\text{training}} = $ "a photo of a {} bird", $t_{\text{unseen}}^1 = $ "a painting of a {} bird".

**DomainNet (multiple new domains)** is a specific split (Tan et al., 2020) of the original DomainNet (Peng et al., 2019) dataset which contains the 40 most common classes from 4 domains: 'sketch', 'real', 'clipart', and 'painting'. Like prior work (Kumar et al., 2022; Tan et al., 2020), we train on sketches and evaluate on the three other domains. We use the domain descriptions $t_{\text{training}} = $ "a sketch of a ", $t_{\text{unseen}}^1 = $ "clipart of a ", $t_{\text{unseen}}^2 = $ "a painting of a ", $t_{\text{unseen}}^3 = $ "a realistic photo of a ".

**Colored MNIST (color bias)** (Arjovsky et al., 2021) was made by taking the original MNIST Digits (Deng, 2012), and coloring them red or blue. In the training and validation sets, even numbers are red and odd numbers are blue, while in the test set digits are colored randomly. The task is to classify the digits $0, 1, .., 9$. We use the domain descriptions "a photo of a red number ", "a photo of a blue number ".

**Waterbirds (contextual bias)** (Sagawa et al., 2019) is a synthetically created dataset which creates contextual bias by taking species of landbirds and waterbirds from the CUB-200 Wah et al. (2011) dataset and pasting them on forest and water backgrounds from the Places (Zhou et al., 2017) dataset. For the training and validation sets, all landbirds appear on forest backgrounds and waterbirds appear on water backgrounds while the test set has an even representation of backgrounds and bird types. We use the domain descriptions "a photo of a {} in the forest", "a photo of a {} on the water".

## 4.3 BASELINES

*Generic and Adaptive zero-shot CLIP* are the zero-shot baselines proposed by Radford et al. (2021): (CLIP ZS (G)) uses the class name alone as the text prompt, while adaptive zero-shot CLIP (CLIP ZS (A)) caters the text prompts to the specific domains (e.g. "a painting of an airplane."). To do well on the extended domain, we average the text embeddings of each class across all possible domains.

*CLIP LP* fits a linear classifier on top of the CLIP image embeddings.

*CLIP LP (ZS init)* initializes the linear classifier with the text embeddings.

*WiSE -LP* (Wortsman et al., 2021) is an ensembling technique which fine-tunes a CLIP model and does a weighted average of the fine-tuned model's weights with the original. Due to the size of the vision and language models we are using, we did not fine-tune the entire backbone and instead ensembled the classifier with the linear classifier probe as explained by Wortsman et al. (2021).

*VQGAN + CLIP* (Crowson et al., 2022) is a method that uses a VQGAN (Esser et al., 2021) trained with CLIP to augment images in pixel space. Using a text prompt and an image, we perform "style transfer" to the new domain in order to augment the training data. We then train a linear probe on the augmented and non-augmented CLIP embeddings. Due to the amount of time and compute required to generate images, we only ran this baseline for DomainNet and augmented approximately 15% of the training dataset. Examples of the augmented images are provided in Table 7.

## 4.4 RESULTS

Table 1 shows in-domain (ID) and out-of-domain (OOD) accuracy on CUB-Paintings and DomainNet. The "Extended" column is the average accuracy of the two, corresponding to the full extended

domain. For CUB-Paintings and DomainNet, *LADS* is able to match or improve the ID accuracy of the fine-tuning baselines while improving over their OOD accuracy. Although CLIP zero-shot achieves higher OOD accuracy on DomainNet, *LADS* achieves the highest result when evaluated on the full extended domain. We also improve over the VQGAN+CLIP baseline on DomainNet.

For Colored MNIST (Figure 3a) and Waterbirds (Figure 3b), *LADS* is able to roughly match ID accuracy of the fine-tuned CLIP and OOD accuracy of CLIP zero-shot, resulting in approximately a 10% improvement on the extended domain. We explore different weighted averages of ID and ODD accuary to compute the extended domain accuracy in Section C of the Appendix.

| Dataset | Method | ID | OOD | Extended |
|---|---|---|---|---|
| CUB-Paintings | CLIP ZS (G) | 60.34% | 52.84% | 56.59% |
| CUB-Paintings | CLIP ZS (A) | 61.93% | 54.38% | 58.16% |
| CUB-Paintings | CLIP LP | **85.91±0.08%** | 64.33±0.29% | 75.12±0.18% |
| CUB-Paintings | CLIP LP (ZS init) | **86.08±0.11%** | 65.05±0.05% | 75.57±0.06% |
| CUB-Paintings | WiSE-LP | 81.74±0.34% | 64.80±0.10% | 73.27±0.22% |
| CUB-Paintings | *LADS* | **86.14±0.29%** | **66.18± 0.25%** | **76.16±0.23%** |
| DomainNet | CLIP ZS (G) | 93.49% | 95.94% | 94.72% |
| DomainNet | CLIP ZS (A) | 93.24% | **96.01%** | 94.62% |
| DomainNet | CLIP LP | 95.03±0.07% | 93.75±0.02% | 94.39±0.04% |
| DomainNet | CLIP LP (ZS init) | **95.21±0.21%** | 93.95±0.03% | 94.58±0.11% |
| DomainNet | WiSE-LP | 95.19± 0.34% | 93.68± 0.12% | 94.44±0.11% |
| DomainNet | VQGAN+CLIP | **95.54± 0.09%** | 93.83± 0.10% | 94.67± 0.09% |
| DomainNet | *LADS* | **95.33 ± 0.33%** | 95.21 ± 0.09% | **95.27± 0.14%** |

Table 1: In-domain (ID), out-of-domain (OOD) and extended domain accuracy on **CUB-Paintings** and **DomainNet**. For DomainNet, we include the pixel augmentation baseline VQGAN+CLIP and OOD accuracy is the average of the 3 unseen domains. *LADS* is able to beat all methods on the extended domain for both datasets. Note that for tasks where CLIP zero-shot does not perform well, *LADS* is able to significantly outperform zero-shot on the unseen domain.

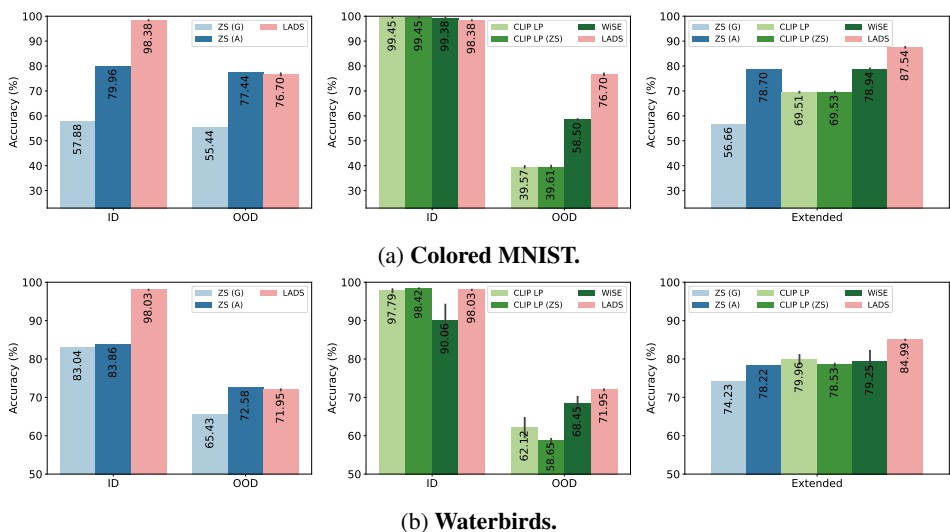

(a) **Colored MNIST.**

(b) **Waterbirds.**

Figure 3: **Result on dataset bias benchmarks.** Left and center plots show the training domain and unseen domain performance of the zeroshot and fine-tuned baselines respectively. For both Colored MNIST (a) and Waterbirds (b), *LADS* is able to roughly match the unseen domain accuracy of zero-shot methods and the seen domain accuracy of fine-tuned methods, resulting in improved performance on the extended domain (right).

|                         | CLIP LP       | VQGAN+CLIP    | *LADS*        |
| ----------------------- | ------------- | ------------- | ------------- |
| Domain Alignment score  | 81.30±1.35%   | 69.26±2.31%   | 85.44±0.61%   |
| Class Consistency score | 91.42±0.47%   | 77.14±1.22%   | 87.26±0.74%   |

Table 2: **Augmentation Quality for DomainNet.** The domain alignment and class consistency scores over 1000 randomly sampled training embeddings and their nearest neighbor in the test set. *LADS* achieves higher domain alignment and class consistency scores than VQGAN+CLIP, and is able to obtain a somewhat similar class consistency score to the unaugmented image embeddings.

## 4.5 ANALYSIS OF AUGMENTATION QUALITY

In this section, we explore the quality of the embeddings generated by our augmentation network. We perform our analysis for DomainNet below, defering the remaining results to Appendix D.1, D.2.

Since there is no publicly available model to convert CLIP embeddings to images, we use the nearest neighbors of the augmented embeddings from the extended test domain to confirm that our augmentations match our expectations. We take a random subset of 1,000 samples from the image embeddings used to train the linear probe: for CLIP LP, this is simply $\{I_\theta(\mathbf{x_i}), y_i\}_{i=1}^n$, for VQGAN+CLIP it is of a mix of $\{I_\theta(\mathbf{x_i}), y_i\}_{i=1}^n$ and GAN generated images, and for *LADS* it is $\{I_\theta(\mathbf{x_i}), y_i\}_{i=1}^n$ and the augmented embeddings $\bigcup_{j=1}^k \{f_{\text{aug}}^j(I_\theta(\mathbf{x_i})), y_i\}_{i=1}^n$ for each unseen domain $j$. We obtain the nearest neighbors in the extended test set (containing images from the training and unseen domain) with respect to cosine similarity of the image embeddings.

In line with our domain alignment and class consistency loss functions, we define metrics for (1) correctly altering the domain of the image embedding, while (2) retaining the class information. We define the percentage of the nearest neighbors that belong in the desired domain as the *domain alignment score*, and the percentage that belong to the original class as the *class consistency score*.

The CLIP LP scores can be viewed as an approximate upperbound for those of *LADS* since they reflect the nearest neighbors of only the original sketch embeddings in the extended domain. As shown in Table 2, *LADS* is able to beat the domain alignment score and closely fall behind the class consistency score of the linear probe, implying that the augmentations are of similar quality to the original image embeddings. Furthermore, *LADS* has better domain alignment and class consistency than VQGAN+CLIP, indicating that the long and laborious pixel-level augmentation may be producing lower quality training samples than our simple embedding augmentation.

For qualitative analysis of *LADS*, we visualize a random sample of 10 nearest neighbors from DomainNet in Figure 4 (the sketch embeddings are non-augmented, all others are augmented). The nearest neighbors of augmented embeddings closely resemble embeddings of similar images in the desired unseen domain. Even if the nearest neighbor is of a different class, it maintains some visual similarity to the original image.

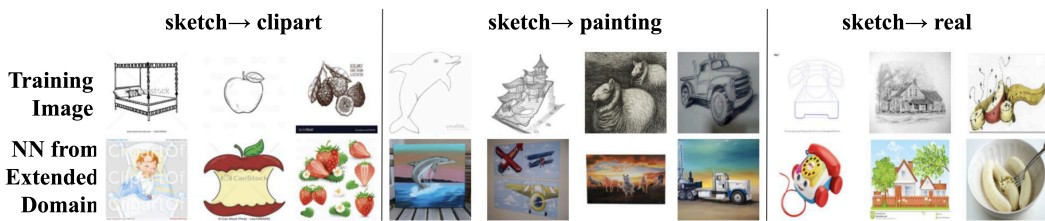

Figure 4: **Nearest Neighbors for *LADS* on DomainNet Sketch → Clipart, Painting, Real.** The top row shows training images with the label on top being the intended domain augmentation for that embedding. The bottom row shows the nearest neighbor of the augmentation in the extended domain. Not only does *LADS* produce augmented embeddings within the correct domain, embeddings often match the specific class and stylistic elements of each original image.

## 4.6 ABLATIONS

In this section, we ablate the class consistency and domain alignment loss described in Section 2. We defer the remainder of the ablations, including ablations of the domain descriptions and the CLIP model, to Appendix D.3, D.4, D.5.

In order to measure the impact of the domain alignment and class consistency loss functions, we ablate each one and report the accuracy, domain alignment score, and class consistency score. We also experiment with a domain-specific class consistency loss, which replaces $T(y_i)$ with $T(t_{\text{unseen}}^k \circ y_i)$ in order to enforce the class and domain all in one loss. We display our results on the Waterbirds dataset below, with experiments on the other datasets in Appendix D.3.

As shown in Figure 5, the domain alignment loss alone results in a high domain alignment score, but low accuracy due to losing some class specific information. Meanwhile, the class consistency loss alone achieves the highest class consistency score because it retains the relevant class information, but it fails to improve the OOD accuracy since the augmented embeddings are not within the new domain. Even in the case of domain specific $\mathcal{L}_{\text{CC}}$ when the extended domain is incorporated into the class consistency loss, the scores only slightly improve. It is only when we combine both our losses that we are able to retain class information while transforming the image embeddings to the desired domain, leading to improved out-of-domain accuracy. Nearest neighbor visualizations of the different losses are given in Appendix D.3.

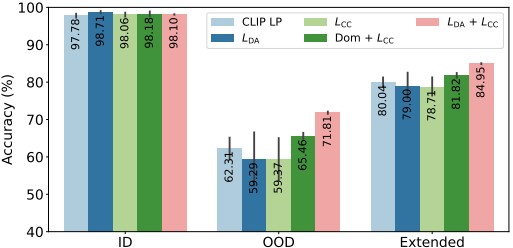 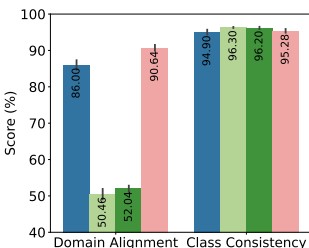

Figure 5: **Effect of the Loss Functions.** We report the results of training with just the domain alignment loss, the class consistency loss, a domain-specific class consistency loss, and the domain alignment + class consistency loss, on Waterbirds. The DA loss results in high DA score but low accuracy. The CC loss results in low DA score and does not improve the OOD accuracy; the domain-specific CC variant brings negligible gains. Our final design ($\mathcal{L}_{\text{DA}}+\mathcal{L}_{\text{CC}}$) works the best.

## 5 LIMITATIONS AND FUTURE WORK

Since one must input a natural language description of the distribution shift, *LADS* may not apply to "natural" distribution shifts where the change cannot be verbalized (Koh et al., 2021). Furthermore, as our approach is reliant on the richness of concepts learned by a pretrained vision-language model, it is also limited to domains that can be accurately represented with textual descriptions, and are well covered in the data the pretrained models were trained on. As a general rule of thumb, if CLIP zero-shot has very poor performance when it comes to classifying the domains and/or classes, *LADS* should not be used (see Section E of the Appendix).

We have presented *LADS*, a fine-tuning method for addressing the task of Domain Extension with Language. We view *LADS* as a jumping-off point for further exploration regarding how we can use the zero-shot capabilities of large multimodal models to improve accuracy on a desired domain given only language description as input. We hope that future work is able to perform reliable embedding augmentations independently of the ability of CLIP to correctly classify the domains and classes at hand. Furthermore, we hope future work is able to analyze more complicated domain shifts such as the ones seen in WILDS Koh et al. (2021) or Imagenet-A (Hendrycks et al., 2021).

**Acknowledgements.** This work was supported in part by DoD including DARPA's SemaFor, PTG and/or LwLL programs, as well as BAIR's industrial alliance programs.

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
