# OpenReview forum: "Using Language to Extend to Unseen Domains"
_ICLR.cc/2023/Conference — ICLR 2023 notable top 25%_

### Official Review · Reviewer_3yhi · 2022-10-17

**Confidence:** 4
**Correctness:** 3
**Technical Novelty And Significance:** 3
**Empirical Novelty And Significance:** 3
**Recommendation:** 8

**Clarity, Quality, Novelty And Reproducibility:**

Quality:

In general, this paper is well-written. The motivation and derivation of proposed techniques are clear to the audience.


Novelty:

This paper shows decent novelty on the studied problem and the methodology used to solve it.


Reproducibility:

Authors did not submit source codes and are recommended to submit them (maybe upon acceptance) for the sake of reproducibility.


**Strength And Weaknesses:**

Strength:

1. This paper studies a practical problem that has been rarely explored, domain extension with language, where the domain gap is filled by utilizing the text descriptions of domains and classes instead of unlabeled target domain samples as in UDA.
2. The proposed method is technically sound. By utilizing pre-trained vision-language alignment models like CLIP, LADS can well represent different domains and classes and further transformer such knowledge to the image modality.
3. Extensive empirical results (performance comparison and ablation study) demonstrate the effectiveness of LADS on tackling domain and dataset shifts.


Weakness:

1. The experimental results under domain shift are not strong enough. For image recognition under domain shift, people are always most interested in the performance on OOD domains. On the two datasets studied in this paper, LADS achieve gains on CUB-Paintings while underperforms zero-short CLIP on DomainNet. Therefore, it is interesting to study under what circumstances LADS is necessary against directly apply the zero-shot CLIP. The current experiments on two datasets are not sufficient to illustrate this. I suggest the authors to do performance comparison on a more diverse set of benchmark tasks under domain shift, e.g., the Office-Home dataset with moderate domain gaps and the VisDA dataset with a larger domain gap.


**Summary Of The Paper:**

This paper proposes the Latent Augmentation using Domain Descriptions (LADS) framework to enhance the generalization ability of image classifiers across various domains. Specifically, LADS trains an augmentation module to transform the source domain image embeddings to multiple pre-known target domains, where domain alignment and class consistency are respectively constrained. After that, a linear classifier is trained with both source and target domain embeddings to achieve better generalization. Under both the settings of domain shift and dataset shift, authors demonstrate that LADS can achieve state-of-the-art performance on the extended domain (the collection of source and target domains).

**Summary Of The Review:**

In summary, I am convinced by the motivation of studying under the proposed setting and believe the technical soundness. I give an acceptance on the current version and suggest authors to do more studies under the domain shift setting during the response period.

---

> ### Author Response · Authors · 2022-11-18
> **Exploring When Zero Shot is Preferred over LADS**
>
> Thank you for your helpful review; we are over-the-moon to hear that you see the practical implications for this work and agree that our empirical results support our claims! \
> We agree that including other DA datasets would give a better understanding of when our method is preferred over zero-shot. However, many DA datasets contain common classes that appear in CLIPs training data (e.g. DomainNet, OfficeHome), so CLIP ZS obtains a high accuracy across all domains and fine-tuning is not necessary. Since we are specifically targeting scenarios where fine-tuning is required, these datasets aren't a good representation of our setting. That being said, we can still examine the results on these datasets to get an idea of when LADS should be used over CLIP ZS. \
> In general, CLIP ZS should be used over LADS when linear probing the CLIP embeddings outperforms ZS on the majority of classes from the source domain. For example, below we have the results for 2 splits of OfficeHome: source domain of clipart and source domain of product, with the test domain being all 4 domains (art, clipart, product, real world). For both splits we see the same phenomenon: since CLIP ZS outperforms CLIP LP on 37/65 classes on the source domain, the overall OOD accuracy of CLIP ZS is higher than LADS. However, if we take the 28/65 classes where CLIP LP outperforms CLIP ZS, then we see that LADS OOD accuracy beats ZS and CLIP LP. \
> We have added a section in the Appendix explaining this phenomenon.
>
>
> | Classes | Method | ID (Clipart) | OOD (A, P, RW) | ID (Product) | OOD (A, C, RW) |
> | ----------- | :-----------: | :-----------: | :-----------:  | :-----------:  | :-----------:  |
> | All | CLIP ZS (G) | 72.07  | 90.19 |  **94.61** | **86.13** |
> | All | CLIP ZS (A) | 74.75  |  **91.34** | 94.49 | 84.73 |
> | All | CLIP LP | **82.66 (0.16)** | 85.95 (0.65) | 95.46 (0.03) | 78.53 (0.40) |
> | All | LADS | **82.55 (0.28)** | 87.97 (0.50) | 96.21 (0.20) | 81.20 (0.22) |
>
> | Classes | Method | ID (Clipart) | OOD (A, P, RW) | ID (Product) | OOD (A, C, RW) |
> | ----------- | :-----------: | :-----------: | :-----------:  | :-----------:  | :-----------:  |
> | ZS Win (37/65) | CLIP ZS (G) | 79.64  | 89.70 | 97.19 | **88.55** |
> | ZS Win (37/65) | CLIP ZS (A) | **80.95**  |  **91.32** | **97.42** | 87.86 |
> | ZS Win (37/65) | CLIP LP | 79.44 (0.17) | 81.92 (0.69) | 95.79 (0.40) | 75.17 (0.62) |
> | ZS Win (37/65) | LADS | 79.15 (0.61) | 84.68 (0.77) | 96.16 (0.25) | 78.79 (0.37) |
>
> | Classes | Method | ID (Clipart) | OOD (A, P, RW) | ID (Product) | OOD (A, C, RW) |
> | ----------- | :-----------: | :-----------: | :-----------:  | :-----------:  | :-----------:  |
> | LP Win (28/65) | CLIP ZS (G) | 62.08  | 90.84 | 90.72 | 82.49 |
> | LP Win (28/65) | CLIP ZS (A) | 66.56  |  91.36  |90.09 | 80.04 |
> | LP Win (28/65) | CLIP LP | **86.91 (0.41)** | 91.28 (0.06) | 94.97 (0.67) | 83.56 (0.09) |
> | LP Win (28/65) | LADS   | **87.20 (0.28)** | **92.31 (0.16)** | **96.29 (0.32)** | **84.82 (0.13)** |

---

### Official Review · Reviewer_AEAE · 2022-10-29

**Confidence:** 3
**Correctness:** 2
**Technical Novelty And Significance:** 3
**Empirical Novelty And Significance:** 2
**Recommendation:** 6

**Clarity, Quality, Novelty And Reproducibility:**

Overall easy to follow. The experiments and analyses support the claims and the contributions for this paper.

**Strength And Weaknesses:**

## Strength

1. The proposed method achieves good performance in both the in-distribution and the out-of-distribution data, without explicitly training on OoD labeled data.

1. The analyses demonstrate that the model indeed successfully learns to transfer the feature into another domain (contribution of $L_{DA}$) and keep the semantic meaning (contribution of $L_{CC}$.

## Weakness

1. The biggest concern the reviewer has is how to describe the unseen domain in the form of simple text. In this paper, the setting is rather simple: there is a clear distinction between training and testing domains, and the difference can be easily described in a few words. As also pointed out in Tab3, the model performance is quite sensitive to the prompt, or domain descriptions.

1. Related to the previous point, in Sec 3.1 the waterbird example, how to discover the additional prior knowledge that the "spurious correlation is the background" may not be trivial.

1. The author should also compare to other more general methods in domain generalization, where only the training domain is available and the domain descriptions of testing domains remain unknown.

1. Since the gap between CLIP-LP and LADS is not that big, how did the authors train the CLIP-LP model? For example, the ELEVATOR paper [1] pointed out that it is beneficial to initialize the clf layer weight by CLIP's language branch.

[1] https://openreview.net/forum?id=hGl8rsmNXzs

**Summary Of The Paper:**

This paper proposed a strategy to augment the image feature into different domains. The process is guided by CLIP to not just transfer to another domain but also keep the semantic content. In the experiment ,we can see that the proposed method, LADS, indeed performs well in both in-distribution and out-of-distribution. We can also qualitatively (Fig4) and quantitatively (Tab2) see that the model indeed learns to transfer the input feature from the training domain to a new domain, and at the same time keep the semantic meaning.

**Summary Of The Review:**

This paper did a good job in the proposed setting, where the unseen domain can be easily described in a few words and is given as prior knowledge during training. However, this seems like a rather contrived setting as described in the weakness section of the review.

---

> ### Author Response · Authors · 2022-11-11
> **Addressing concerns around the use of language.**
>
> Thank you for your thorough review of our paper, we address your concerns around the use of language in the problem setup.
>
> **The ability to summarize the unseen domain seems unrealistic.**
>
> As our paper is centered around how language can be used to improve unseen domain accuracy, we agree that LADS is limited by the users ability to verbalize the source and target domain. However, as described in our motivating example (Section 1), we believe that this setting does appear often, as domain shifts based on weather, time of day, or background can usually be described with language. Most domain adaptation datasets actually fit into this setup; a fact which has not been fully utilized in previous work.
>
> That being said, we do think that it would be interesting in future work to apply this idea of augmenting embeddings on more subtle domain shifts such as WILDS. One simple thing that someone could try is replacing the text embeddings of the source and target domains with images of the source and target domain. For example, one could take the iWildCam dataset where the domain shift is the camera location, and use images of each location with no animals in it. Then the same training method as LADS could be applied, with the direction vector being the difference in these image embeddings. Since this method does not utilize language of the domains it is beyond the scope of our work, but an interesting avenue nonetheless.
>
> **As also pointed out in Tab3, the model performance is quite sensitive to the prompt, or domain descriptions.**
>
> Given the premise of being able to describe the training and unseen domains, we believe that our method’s sensitivity to prompts is confirmation that it is actually incorporating external knowledge about the text descriptions of the unseen domains. As Table 3 shows, two different wordings describing the correct domain shifts achieve similar results, while the prompts with incorrect or gibberish descriptions of the training and unseen domains have significantly worse OOD and Extended domain performance. This difference ensures that LADS leverages the text information of domain shifts to extend the domain, and rules out the possibility that the augmentation is only adding random noise to boost generalization.
>
> **In Sec 3.1 (the waterbird example), how to discover the additional prior knowledge that the "spurious correlation is the background" may not be trivial.**
>
> While many previous debiasing methods[1,2,3,4] also make the assumption that the spurious correlation is known, we agree that this is a weakness of our method similar to the requirement of a language description of the domain shift. One thing that could be done to find spurious correlations is to create a list of attributes that are known to appear in the dataset and run CLIP ZS to get per-image attribute labels. Then one can see which attributes overwhelmingly appear with a specific class/set of classes to hypothesize spurious correlations.
>
> [1] Kim et a;. “Training deep neural networks with biased data“
>
> [2] Adeli et al. “Representation learning with statistical independence to mitigate bias”
>
> [3] Kim et al. “Learning not to learn: Training deep neural networks with biased data.”
>
> [4] Sagawa et al. “Distributionally robust neural networks for group shifts: On the importance of regularization for worst- case generalization.”

---

> ### Author Response · Authors · 2022-11-11
> **Additional baselines.**
>
> We address your suggestions to incorporate other baselines to compare LADS to.
>
> **The author should also compare to other more general methods in domain generalization, where only the training domain is available and the domain descriptions of testing domains remain unknown.**
>
> We would be happy to compare to other DG methods! To our knowledge, the DG setup assumes multiple source domains, so methods ([5,6,7]) require more than one source domain to operate. If you have a particular DG method in mind that can work with one source domain, please let us know and we will happily add it as a baseline before the rebuttal period ends, time permitting.
>
> [5] Arjovsky et al. “Invariant Risk Minimization” \
> [6] Zhao et al. “Domain Generalization via Entropy Regularization” \
> [7] Muandet et al. “Domain Generalization via Invariant Feature Representation“
>
> **Since the gap between CLIP-LP and LADS is not that big, how did the authors train the CLIP-LP model? For example, the ELEVATOR paper [1] pointed out that it is beneficial to initialize the clf layer weight by CLIP's language branch.**
>
> Thank you for bringing this baseline to our attention! We trained a single layer MLP with a random initialization, but initializing the weights to the text embeddings is certainly another applicable baseline. \
> Below we have the results for LP with the text initialization (we use the adaptive prompts we use for the CLIP ZS (A) baseline) and a bias initialized to zero as described in [9]. We perform the same hyperparameter sweep to this method as our other methods described in Section A of the Appendix and results are over 5 random seeds. We include the results of CUB and Waterbirds below, and will update the main results for all datasets in the manuscript. While initializing the weights to the text embeddings does improve accuracy over a random initialization for CUB, LADS is able to achieve higher OOD and Extended domain accuracy.
>
> [9] Li et al. “ELEVATER: A Benchmark and Toolkit for Evaluating Language-Augmented Visual Models”
>
>
> | Dataset      	       | Method | ID	Acc (%) | OOD Acc (%)	 | Extended Acc (%) |
> | :---------- | :----------- | :-----------: | :-----------: | :-----------: |
> | CUB-Paintings   | CLIP LP       	| **85.91 (0.08)** | 64.33 (0.29) | 75.12 (0.18) |
> | CUB-Paintings   | CLIP LP (ZS init)| **86.08 (0.11)** | 65.05 (0.05) | 75.57 (0.06) |
> | CUB-Paintings   | LADS        	| **86.14 (0.29)** | **66.18 (0.25)** | **76.16 (0.23)** |
>
> | Dataset      	       | Method | ID	Acc (%) | OOD Acc (%)	 | Extended Acc (%) |
> | :---------- | :----------- | :-----------: | :-----------: | :-----------: |
> | Waterbirds	| CLIP LP	            | 97.79 (0.53)	| 62.12 (3.47) | 79.96 (1.49) |
> | Waterbirds	| CLIP LP (ZS init)  | **98.41 (0.01)** | 58.65 (1.00) | 78.53 (0.50) |
> | Waterbirds	| LADS        		 | 98.02 (0.06) | **71.95 (0.18)** | **84.99 (0.07)** |

---

### Official Review · Reviewer_hq6S · 2022-11-02

**Confidence:** 4
**Correctness:** 4
**Technical Novelty And Significance:** 3
**Empirical Novelty And Significance:** Not applicable
**Recommendation:** 6

**Clarity, Quality, Novelty And Reproducibility:**

The paper addresses a novel problem. The paper is simple to understand. The paper provides setting details that support reproducibility.

**Strength And Weaknesses:**

(Strengths)

- The proposed method does not require any unseen domain image samples. It significantly reduces the cost of data collection.

- LADS augments data at the image space level rather than the pixel level. This approach is useful for avoiding bottlenecks caused by the generative process's quality.

- The proposed method can be extended to address training data biases.

- Extensive experiments are carried out to demonstrate the performance of LADS in comparison to the state-of-the-art. Furthermore, the authors discuss the proposed method's limitations on the 'natural' distribution shift.

(Weaknesses)

- The text embeddings of the domain descriptions guide the domain alignment loss. During training, the image embeddings space changes while the text embeddings space remains constant. It is suspected that the text embeddings function properly as guidance.
- Furthermore, the method is heavily reliant on the domain descriptions used. Taking the first two rows of table 3, the results are affected by the prompts. Considering the variations of the 'direction' instead of fixing a 'global direction' may be helpful.
- Though the ablation study investigated the role of each loss term, it is suggested that the role of $\alpha$ be investigated as well.


**Summary Of The Paper:**

The problem of domain extension with language is addressed in this paper. The proposed method (LADS) uses a CLIP model's domain-level knowledge to learn a latent feature augmentation of the training set. It does not require any unseen domain samples and instead relies on written descriptions of the training and unseen domains. Domain alignment and class consistency losses are used to train the latent feature augmentation. Once trained, a simple linear classifier is trained on both the original and augmented image embeddings, resulting in improved in-domain and out-of-domain recognition performance. Experiments on DomainNet, CUB-Paintings, Colored MNIST, and Waterbirds demonstrate that LADS outperforms other methods.

**Summary Of The Review:**

The paper's contributions are novel. The technique is simple, but it works well. However, it is suggested that more research into domain alignment loss is required. Given the novelty of the paper, I would recommend a 'marginally above the acceptance threshold' rating.

---

> ### Author Response · Authors · 2022-11-11
> **Addressing the concerns around a "global direction" and the role of $\alpha$.**
>
> We appreciate your thoughtful review and are excited that you see the novelty in our work. We address your comments below.
>
> **The method is heavily reliant on the domain descriptions used. Taking the first two rows of table 3, the results are affected by the prompts. Considering the variations of the 'direction' instead of fixing a 'global direction' may be helpful.**
>
> Our method's sensitivity to prompts is confirmation that our method is in fact incorporating external knowledge instead of adding random noise which boosts generalization. Table 3 shows that similar prompts have similar results while nonsense prompts have poor results, so two comparable descriptions of the source and target domain should lead to roughly the same results.
>
> That being said, considering variations of the direction is a great idea! We are currently running experiments which have a set of directions created by enumerating the differences of several valid descriptions of the source and target domain (i.e. [“a photo of a bird”, “an image of a bird”] and [“a painting of a bird”, “an artistic rendition of a bird”]). We choose which direction(s) to use in the domain alignment loss of a particular image based on which source domain description has the highest similarity to the unaugmented image embedding. The results for this experiment will be posted before the rebuttal period ends.
>
> Please let us know if you had another experiment in mind; we would be happy to try it out.
>
> **Though the ablation study investigated the role of each loss term, it is suggested that the role of $\alpha$ be investigated as well.**
>
> To clarify, our loss ablation is equivalent to setting $ \alpha = 0$ (only $L_{CC}$) or $ \alpha =1$ (only $L_{DA}$), but we agree that a more comprehensive analysis would be insightful. Below we have included performance of LADS on CUB for $\alpha$ values 0, 0.25, 0.75, and 1.0. Note that the domain alignment and class consistency scores are the proportion of nearest neighbors of the augmented embeddings that match the desired (target) domain and class.
>
> As shown in our loss ablation, when  $ \alpha = 0$, the class consistency is high and the domain alignment is low, while the opposite is seen when  $ \alpha = 1$. However, we see that we can often achieve a better domain alignment score when $\alpha$ is between 0 and 1. We believe that this is because the augmented embeddings drift too far from the unaugmented embeddings without the CLIP supervision that the class consistency score provides.
>
> We will update our manuscript to also include the loss ablation for Waterbirds.
>
> | Dataset      | $\alpha$ | DA Score (%) | CC Score (%)	 | ID	Acc (%) | OOD Acc (%)	 | Extended Acc (%) |
> | :----------- | :-----------: | :-----------: | :-----------: | :-----------: |:-----------: | :-----------: |
> | CUB-Paintings      | 0.0      |9.5 (0.95) | 57.46 (3.21) | 85.88 (0.14) | 64.59 (0.41) | 75.24 (0.27) |
> | CUB-Paintings   | 0.25      |89.16 (3.21) | 51.28 (5.93) | 85.89 (0.17) | 65.04 (0.73) | 75.47 (0.45) |
> | CUB-Paintings   | 0.5      | 99.64 (0.27) | 54.84 (2.69) | 86.14 (0.29) | 66.18 (0.25) | 76.16 (0.23) |
> | CUB-Paintings   | 0.75      | 94.20 (1.00) | 44.56 (2.27) | 85.63 (0.09) | 62.93 (0.31)  | 74.28 (0.18) |
> | CUB-Paintings   | 1.0      | 89.36 (1.60) | 37.48 (2.03) | 83.87 (0.15) | 53.74 (0.76) | 68.80 (0.41) |

---

> ### Author Response · Authors · 2022-11-17
> **Results using multiple source domain descriptions**
>
> **Update:** below are the results for DomainNet with the source prompts: ['a sketch of a {}', 'a pencil drawing of a {}.', 'a drawing of a {}.'] and the same target prompts as described in the experimental setup of the paper. It seems like extended accuracy is roughly the same when using more source domains, but we will certainly explore this idea more after the rebuttal period.
>
> | Dataset  | Method 	| ID | OOD  | Extended |
> | :----------- | :-----------: |  :-----------:  | :-----------:  | :-----------:  |
> | DomainNet   | CLIP LP       	| 95.03 (0.07) | 93.75 (0.02) | 94.39 (0.04) |
> | DomainNet  | LADS        	| **95.33 (0.33)** | **95.21 (0.09)** | **95.27 (0.14)** |
> | DomainNet   | LADS (multi-source)   | **95.53 (0.04)** | **95.20 (0.05)** | **95.37 (0.03)** |

---

### Decision · Program_Chairs · 2023-01-20

**Decision:**

Accept: notable-top-25%

**Justification For Why Not Higher Score:**

All reviewers basically acknowledge the novelty of the addressed problem and the proposed method as well as the appropriateness of the experimental validation. They are unanimous for the acceptance and the AC also agrees that the paper is well above the acceptance threshold. Meanwhile, the method has some obvious limitations due to some common problems in this kind of tasks (e.g., how to appropriately setup and utilize prior knowledge for linguistic guidance). From a practical viewpoint, the method is a little better than baseline CLIP-based probing but not a very big jump. In this sense, the method is good but not ground-breaking, so the AC believes it is a little short for oral presentation.

**Justification For Why Not Lower Score:**

That being said (above), the paper does show a first concrete step showing the new direction in domain adaptation, i.e., language-guided domain extension. It would be valuable for the community by widely spreading its core idea in a spotlight.

**Metareview: Summary, Strengths And Weaknesses:**

Summary:
This paper explores a language-based domain extension strategy. The proposed method, LADS, transforms the source domain image embeddings to multiple pre-known target domains guided by CLIP model’s knowledge. Experiments on several datasets demonstrate that LADS improves the performance on both in-domain and out-of-domain recognition settings, and outperforms other methods.

Strengths:
1. This paper addresses a rarely explored problem of language-guided domain extension which is well motivated and thought to be practically useful.
2. The proposed method is technically sound and solid.
3. The proposed method achieves good performance in both the in-distribution and the out-of-distribution data, without explicitly training on OoD labeled data.
4. Analysis and discussion are good and convincing that the model indeed successfully learns to transfer the feature into another domain and keep the semantic meaning.
5. During the discussion period, the authors added new results showing that the proposed method is indeed better than simple linear probing of CLIP.

Weaknesses:
1. The method is based on the assumption that the unseen domain description is given and can be expressed in the form of relatively simple text. It is unclear how the method can be applied to more general cases. Also, debiasing spurious correlation would require further prior knowledge.



**Note From Pc:**

if the above contains the word "oral" or "spotlight" please see: "oral" presentation means -> notable-top-5% and "spotlight" means -> notable-top-25%. As stated in our emails, we are disassociating presentation type from AC recommendations